# Interplay between Lung Diseases and Viral Infections: A Comprehensive Review

**DOI:** 10.3390/microorganisms12102030

**Published:** 2024-10-08

**Authors:** Chahat Suri, Babita Pande, Lakkakula Suhasini Sahithi, Tarun Sahu, Henu Kumar Verma

**Affiliations:** 1Department of Oncology, Cross Cancer Institute, University of Alberta, Edmonton, AB T6G 1Z2, Canada;csuri@ualberta.ca; 2Department of Physiology, All India Institute of Medical Sciences, Raipur 492099, India; babitatime2014@gmail.com (B.P.);tarunsahu@aiimsraipur.edu.in (T.S.); 3Department of Biotechnology, Guru Ghasidas Vishwavidyalaya, Bilaspur 495009, India; lakkakulasahithi@gmail.com; 4Department of Immunopathology, Institute of Lungs Health and Immunity, Comprehensive Pneumology Center, Helmholtz Zentrum, Neuherberg, 85764 Munich, Germany

**Keywords:** lung disease, viral infection, interaction, exacerbation, pathway, therapeutics

## Abstract

The intricate relationship between chronic lung diseases and viral infections is a significant concern in respiratory medicine. We explore how pre-existing lung conditions, including chronic obstructive pulmonary disease, asthma, and interstitial lung diseases, influence susceptibility, severity, and outcomes of viral infections. We also examine how viral infections exacerbate and accelerate the progression of lung disease by disrupting immune responses and triggering inflammatory pathways. By summarizing current evidence, this review highlights the bidirectional nature of these interactions, where underlying lung diseasesincrease vulnerability to viral infections, while these infections, in turn, worsen the clinical course. This review underscores the importance of preventive measures, such as vaccination, early detection, and targeted therapies, to mitigate adverse outcomes in patients with chronic lung conditions. The insights provided aim to inform clinical strategies that can improve patient management and reduce the burden of chronic lung diseases exacerbated by viral infections.

## 1. Introduction

Chronic lung diseases (CLDs), including chronic obstructive pulmonary disease (COPD), asthma, and interstitial lung diseases (ILDs), represent significant challenges in respiratory medicine, impairing pulmonary function and increasing vulnerability to viral infections. These conditions not only exacerbate respiratory distress but also demand a comprehensive understanding of their intricate relationship with viral infections [1]. Notably, about half of COPD exacerbations are triggered by bacterial and viral infections, particularly rhinovirus, and air pollution is another prominent contributor to exacerbations [2]. Exacerbations, occurring more frequently than previously recognized, are often managed in the community without hospital admission, and approximately half of them go unreported by patients [3]. Early physiologic changes, such as decreases in peak flow and forced expiratory volume in the first second (FEV1), are generally small and thus not predictive of exacerbations. However, more significant decreases in peak flow are associated with dyspnoea and symptomatic upper respiratory viral infections [4].

CLDs, marked by persistent respiratory symptoms and airflow limitation, create an environment conducive to viral infections. The compromised respiratory defences and structural abnormalities associated with COPD, asthma, and ILDs increase susceptibility to respiratory viruses [5]. Impaired mucociliary clearance, altered immune responses, and chronic airway inflammation further facilitate viral replication and dissemination [6]. Consequently, individuals with CLDs are not only more prone to viral infections but also experience more severe illness and prolonged recovery, leading to increased morbidity and mortality [7]. Viral infections, in turn, can exacerbate the progression of chronic lung diseases, precipitating exacerbations and accelerating disease deterioration [8,9]. The inflammatory response triggered by viral pathogens can worsen airway inflammation, bronchoconstriction, and mucus hypersecretion, further compromising respiratory function in these patients [10].

The airway epithelium serves as the first line of defence against viral infections, maintaining barrier integrity, secreting inflammatory mediators, and initiating an antiviral response through interferon (IFN) production [11]. IFNs play a crucial role in host defence by activating signalling cascades that combat viral infections. However, CLDs often exhibit an altered antiviral response [12]. Exacerbations are frequently associated with bacterial pathogens, particularly *Haemophilus influenzae*, which is found in 69.6% of cases, while rhinovirus is identified in 19.6% [13]. Although rhinovirus does not cause airway epithelial damage like influenza and other respiratory viruses, it can impair the innate immunity of the airway epithelium, exacerbating the condition [14]. These viral-induced exacerbations not only pose immediate threats but also contribute to the long-term decline in lung function and quality of life [15].

Understanding the bidirectional relationship between CLDs and viral infections requires elucidating the underlying mechanisms driving these interactions [16]. Dysregulated immune responses, characterized by impaired antiviral defences and exaggerated inflammation, are central to the pathogenesis of both CLDs and viral infections [17]. Complex signalling pathways involving cytokines, chemokines, and immune cells mediate the interaction between viral pathogens and pulmonary tissues, influencing disease outcomes [18]. Therefore, a holistic approach to disease management is essential [19]. Preventative measures, such as vaccination against common respiratory viruses and strict infection control, are vital to reducing the risk of viral infections in patients with CLDs [20]. Early detection and prompt antiviral treatment can mitigate disease severity and improve clinical outcomes [21]. Integrated treatment strategies that address both the underlying lung disease and associated viral complications are crucial for optimizing patient care and enhancing respiratory health [22].

## 2. Impact of Viral Infections on Chronic Lung Diseases

Viral infections exert a profound impact on the pathogenesis and clinical course of CLDs, representing significant triggers for exacerbations and disease progression [1,23]. Chronic lung diseases, including COPD, asthma, ILDs, and pulmonary hypertension (PH), are characterized by persistent respiratory symptoms and airflow limitations, leading to impaired pulmonary function and diminished quality of life [24]. Viral pathogens, ranging from influenza viruses and respiratory syncytial virus (RSV) to rhinoviruses and coronaviruses, can infect the respiratory tract, initiating a cascade of inflammatory and immune responses that exacerbate underlying lung pathology [25]. Figure 1 depicts the overview of the viral infection pathway and consequences in CLDs like asthma, COPD, and ILD.

### 2.1. Chronic Obstructive Pulmonary Disease (COPD)

Viral infections are agents that amplify the exacerbations in individuals with COPD, primarily by enhancing inflammatory responses and impairing airway clearance mechanisms [26]. Respiratory viruses, such as influenza, respiratory syncytial virus (RSV), and rhinoviruses, infect the respiratory epithelium and initiate a cascade of inflammatory processes [27]. Upon viral invasion, the innate immune system activates, leading to the release of pro-inflammatory cytokines, chemokines, and other mediators, which exacerbate airway inflammation and the chronic inflammation characteristic of COPD [28]. Additionally, viral infections recruit and activate neutrophils, macrophages, and other immune cells, further intensifying the airway inflammation [29].

Viral infections also impair mucociliary clearance, a critical defence mechanism, leading to mucus hypersecretion and airway obstruction, worsening airflow limitation and respiratory symptoms in COPD patients [30]. Damage to the respiratory epithelium disrupts the airway barrier, promoting bacterial colonization and secondary infections that further exacerbate inflammation and impair lung function [31]. Understanding these mechanisms is crucial for developing targeted therapies to mitigate the impact of viral infections on COPD exacerbations [32].

### 2.2. Asthma

Viral infections, particularly rhinoviruses, are key triggers of severe asthma exacerbations, underscoring the complex interaction between viral pathogens and asthma pathophysiology [33]. Rhinoviruses, which predominantly infect the respiratory epithelium, are the most common cause of asthma exacerbations, responsible for a significant proportion of acute respiratory illnesses in asthmatic individuals [34]. The pathogenesis of these exacerbations involves intricate interactions between viral pathogens and the host immune response [35]. Upon infection, rhinoviruses activate pattern recognition receptors (PRRs) on respiratory epithelial cells, initiating an innate immune response characterized by the release of pro-inflammatory cytokines, chemokines, and type I interferons [36].

Key cytokines such as interleukin (IL)-6, IL-8, and tumour necrosis factor-alpha (TNF-α) play critical roles in driving airway inflammation and bronchial hyperreactivity [37]. IL-6 promotes the recruitment of inflammatory cells, including neutrophils and eosinophils, and supports Th17 cell differentiation, both contributing to airway inflammation [38]. IL-8, a potent chemokine, attracts neutrophils and monocytes, and its elevated levels correlate with disease severity in asthmatic patients [39]. TNF-α, produced by activated macrophages, amplifies the inflammatory cascade and contributes to airway remodelling, smooth muscle contraction, and mucus hypersecretion, worsening bronchoconstriction and respiratory symptoms [40].

Overall, viral infections, particularly those caused by rhinoviruses, represent a major precipitating factor for severe asthma attacks, primarily through their ability to induce airway hyperresponsiveness and inflammation [41]. Understanding the mechanisms underlying viral-induced asthma exacerbations is crucial for developing targeted therapeutic interventions aimed at mitigating the impact of viral infections on disease severity and improving clinical outcomes in asthmatic individuals [42]. Furthermore, viral infections can disrupt epithelial barrier functions, compromising the integrity of the airway epithelium and increasing the permeability of the mucosal surface [43]. This disruption allows for enhanced allergen exposure and sensitization, exacerbating allergic airway inflammation and asthma symptoms [44]. Moreover, viruses may promote airway remodelling processes, including subepithelial fibrosis and smooth muscle hypertrophy, further contributing to airway dysfunction and asthma severity [45].

### 2.3. Interstitial Lung Diseases (ILDs)

Viral infections can exacerbate ILDs by accelerating fibrotic processes and worsening lung function [46]. ILDs encompass a diverse group of parenchymal lung disorders characterized by inflammation and fibrosis of the lung interstitium [47,48]. Viral pathogens, including influenza viruses, herpesviruses, and adenoviruses, are implicated in the pathogenesis and progression of ILDs, especially in susceptible individuals. Upon viral infection, the host immune response triggers the release of pro-inflammatory cytokines and chemokines, promoting immune cell recruitment and activation within the lung interstitium [49]. This inflammatory environment fosters fibrotic processes, marked by excessive deposition of extracellular matrix components, such as collagen and fibronectin. Viral-induced inflammation also stimulates fibroblasts, key effector cells in tissue fibrosis, further perpetuating the fibrotic response [50].

The impact of COVID-19 on ILDs has also been explored, especially the development of pulmonary fibrosis [51].In SARS-CoV-2 infection, the virus’s spike protein binds to angiotensin-converting enzyme (ACE)-2 receptors on host cells. ACE converts angiotensin (Ang) I to Ang II, which has pro-inflammatory and pro-fibrotic properties through the activation of signalling pathways like Transforming Growth Factor-Beta (TGF-β), Interleukin-1β, TNF-α, and IL-6 [52]. The resulting cytokine storm—an excessive immune response—leads to pulmonary interstitial edema and severe inflammation, contributing to acute respiratory distress syndrome (ARDS) [53].

Viral infections may also directly contribute to lung fibrosis through mechanisms such as cytotoxic effects and induction of the epithelial-mesenchymal transition (EMT) [54]. Viruses infect alveolar epithelial cells, causing cell death and disrupting alveolar integrity [55]. They may also trigger the EMT, where epithelial cells acquire a mesenchymal phenotype, contributing to the pool of activated fibroblasts involved in tissue remodelling and fibrosis [56]. Additionally, viral-induced cytokine storms exacerbate pulmonary inflammation and fibrosis in ILD patients [57]. These cytokine cascades perpetuate tissue injury and remodelling, leading to progressive lung function deterioration [58].

### 2.4. Pulmonary Hypertension (PH)

Viral infections also play a significant role in the pathogenesis and progression of PH, exacerbating vascular dysfunction and increasing pulmonary arterial pressures [59]. The mechanisms behind viral-induced PH are complex, but studies have shed light on how viral pathogens contribute to pulmonary vascular remodelling [60,61]. Influenza viruses, for example, are implicated in the development of PH, particularly in individuals with pre-existing cardiovascular or respiratory conditions [62]. Research has shown a significant association between influenza infection and acute cardiovascular events, such as myocardial infarction and ischemic stroke, underscoring the potential impact of viral infections on vascular health [63]. In animal models, influenza-induced inflammatory responses have been shown to promote endothelial dysfunction and vascular remodelling, leading to increased pulmonary vascular resistance and elevated pulmonary arterial pressures [64].

PH is a common complication of COPD and is associated with worsening clinical outcomes, including more frequent exacerbations and increased healthcare utilization. PH typically progresses slowly but can lead to significant morbidity, even without altering right ventricular function in most cases [65]. Coronaviruses, such as SARS-CoV-2 and Middle East respiratory syndrome coronavirus (MERS-CoV), are also known for inducing severe respiratory illness and systemic inflammatory responses [66]. Studies have demonstrated that coronavirus infections can directly damage endothelial cells and vascular smooth muscle cells, leading to endothelial dysfunction and pulmonary vascular remodelling [67]. Coronavirus-induced cytokine storms further exacerbate vascular inflammation and thrombotic complications, increasing the risk of PH [68].

Adenoviruses, common respiratory pathogens, are also implicated in PH pathogenesis. They activate inflammatory pathways, including nuclear factor-kappa B (NF-κB) signalling and toll-like receptor (TLR) activation, leading to the release of pro-inflammatory cytokines and chemokines in the pulmonary vasculature [69]. Adenoviruses have been shown to disrupt vascular endothelial growth factor (VEGF) signalling pathways, contributing to pulmonary vascular remodelling and PH [70]. Viral infections, including those caused by influenza viruses, coronaviruses, and adenoviruses, indirectly affect vascular smooth muscle and endothelial function in the lungs, leading to or worsening PH [71]. Viral-induced endothelial dysfunction and vascular remodelling compromise pulmonary vascular integrity, increasing susceptibility to vasoconstrictive stimuli and impairing vasodilatory responses [72]. Additionally, viral pathogens can disrupt endothelial barrier functions, promoting vascular leakage and the extravasation of inflammatory cells and cytokines into the pulmonary interstitium [52]. Viral-induced oxidative stress and cellular damage further exacerbate vascular dysfunction, contributing to the pathogenesis of PH [73]. In conclusion, viral infections significantly impact the progression of ILDs and PH by accelerating fibrosis, promoting vascular dysfunction, and exacerbating inflammatory responses. Understanding these mechanisms is crucial for developing targeted therapeutic strategies to mitigate the impact of viral infections on lung diseases.

## 3. Mechanisms of Disease Exacerbation

Viral infections exacerbate CLDs by intensifying inflammation, disrupting immune responses, and worsening lung function. CLDs, which include conditions like COPD, asthma, and ILDs, pose significant challenges due to their impact on respiratory health. Viral infections in patients with CLDs lead to increased inflammation and tissue damage, disturbing the delicate immune balance in the lungs. This disruption results in elevated levels of pro-inflammatory cytokines and compromised antiviral defences, further exacerbating CLD [74].

In COPD, for instance, exacerbations are often marked by increased serum IL-6 and plasma fibrinogen levels, correlating with acute flare-ups [75]. Air pollutants, particularly cigarette smoke (CS), are major contributors to inflammatory processes in COPD. When inhaled, pollutants attract and activate inflammatory cells in the airway mucosa, including CD8 T cells, CD4 helper T cells, and neutrophils, leading to sustained immune responses. These responses contribute to elastin degradation in the alveoli, resulting in emphysema and loss of lung elasticity. Additionally, neutrophil activation leads to mucus hypersecretion from goblet cells, further worsening airflow obstruction [76].

Moreover, macrophages and epithelial cells secrete transforming growth factor-β (TGF-β), which promotes fibroblast proliferation. Fibroblasts, in turn, secrete connective tissue growth factor, contributing to lung epithelial senescence and decreased cellular regeneration, which are key factors in emphysema development. Granzyme B, secreted by CD8+ cells, also plays a role in extracellular matrix degradation and tissue remodelling associated with emphysema [77].

### 3.1. Immune Dysregulation in CLDs

CLDs are associated with significant immune dysregulation, impairing the host’s ability to effectively combat viral pathogens. In CLD patients, alterations in immune cell populations, such as macrophages, dendritic cells, and T lymphocytes, contribute to immune dysfunction and weakened antiviral defence mechanisms [78]. Upon viral infection, respiratory epithelial cells are primary targets for viral replication, triggering innate and adaptive immune responses aimed at controlling the spread of the virus [79]. However, in individuals with CLDs, compromised epithelial barriers, impaired mucociliary clearance, and dysregulated cytokine signalling create an environment that facilitates viral replication and propagation [80]. This dysregulation of immune responses leads to poor outcomes and increased susceptibility to secondary bacterial infections [81].

Respiratory viruses, such as rhinoviruses and influenza viruses, exacerbate airway inflammation in CLD patients by stimulating the release of pro-inflammatory cytokines, chemokines, and type I interferons. These inflammatory mediators worsen underlying lung pathology and respiratory symptoms [82]. Additionally, viral-induced immune dysregulation impairs the function of innate immune cells, including neutrophils, macrophages, and natural killer cells, further compromising the host’s defence mechanisms and exacerbating respiratory symptoms [83].

### 3.2. Inflammatory Pathways and Viral Exacerbation of CLD

Specific inflammatory mediators and cells play critical roles in exacerbating lung disease following viral infection [84]. In COPD patients, viral-induced exacerbations are characterized by heightened airway inflammation, driven by elevated levels of pro-inflammatory cytokines such as IL-6, IL-8, and TNF-α. These cytokines promote the recruitment and activation of neutrophils and macrophages within the airways, intensifying bronchial inflammation and airflow limitation [85,86]. Similarly, in asthma, viral infections trigger robust immune responses characterized by eosinophilic inflammation and the production of Th2 cytokines, including IL-4, IL-5, and IL-13. These cytokines contribute to airway hyperresponsiveness, mucus hypersecretion, and bronchoconstriction, exacerbating respiratory symptoms and leading to severe asthma attacks [87,88]. The activation of mast cells and basophils during viral infections further amplifies the inflammatory cascade in the airways, worsening bronchial inflammation and airway obstruction [89].

In ILDs, viral pathogens stimulate the release of pro-fibrotic cytokines, including TGF-β and platelet-derived growth factor (PDGF), which promote the activation and differentiation of fibroblasts into myofibroblasts. This process results in excessive collagen deposition and pulmonary fibrosis, significantly impairing lung function [90,91].

## 4. Impact of Cigarette Smoke and Viral Infections on Inflammatory Responses

Cigarette smoke exposure further exacerbates the inflammatory processes in CLD by damaging the airway epithelium and releasing pro-inflammatory mediators. This damage triggers the release of damage-associated molecular patterns (DAMPs) and heightens the pro-inflammatory cascade. Macrophages upregulate IL-33 and downregulate innate type-2 lymphoid cells, making the airway more susceptible to viral infections and bacterial dysbiosis due to compromised barrier function [92,93]. The combined effects of these events lead to an inflammatory cascade favouring the activation of innate lymphoid type 2 cells (ILC2) over ILC1 cells. This results in the release of inflammatory mediators and cytokines, including IL-1α, IL-1β, IL-33, and IL-18, through the activation of the NLRP3 inflammasome [94].

In response to this inflammation, the adaptive immune system polarizes towards a Th1 and Th17 response, with the recruitment of CD4+ T helper type 1 (Th1) and type 17 (Th17) cells, producing IFN-gamma and IL-17A, respectively. Severe infections, particularly in the reduced airways, are marked by the accumulation of B cells, T cells, and follicular dendritic cells [95]. In the context of severe viral infections like COVID-19, pathways involving inflammatory cytokines and chemokines, such as IL-6, IL-1β, and TNF-α, are significantly elevated, contributing to cytokine release syndrome (CRS) [96]. IL-6, a key pro-inflammatory cytokine, is crucial for recovery from multiple viral infections but can be detrimental if overproduced, as it hinders viral clearance and survival [97].

## 5. IL-6 as a Biomarker and Therapeutic Target

As we have discussed earlier, SARS-CoV-2 infection or COVID-19 impacts the condition of chronic lung diseases such as ILDs and PH. IL-6 levels have been closely associated with the severity of pulmonary disease in COVID-19 patients. For instance, serum IL-6 concentrations correlated with disease severity in a cohort of 69 hospitalized SARS-CoV-2 patients, while other cytokines, like IL-2 and IL-4, did not show similar correlations [98]. A meta-analysis of 25 COVID-19 studies reported an average IL-6 concentration of approximately 36.7 pg/mL [99]. Although a direct correlation between viral load and IL-6 production is not well established, COVID-19 patients exhibit a direct or indirect relationship with the viral infection. Unlike CRS patients following CAR-T cell therapy, who did not show underlying viral replication, COVID-19 patients experienced prolonged IL-6 elevations [100].

As a result, IL-6 may serve as both a biomarker and a therapeutic target for COVID-19, particularly in the absence of direct-acting antiviral treatments. Emerging clinical cut-off values for IL-6 are being proposed to guide therapeutic interventions in COVID-19 patients [101]. IL-6 peaks are temporary in CAR-T patients, particularly if therapy is in place, but IL-6 levels persist in COVID-19 patients for an extended period of time [102].IL-6 clinical cut-off values that are recommended in this context are beginning to appear, despite the small sample size [38,78].

Overall, SARS-CoV-2 viral infections significantly worsen CLD by intensifying inflammation, disrupting immune responses, and exacerbating underlying lung pathology. Understanding the mechanisms of viral-induced exacerbations and the role of inflammatory mediators like IL-6 is crucial for developing targeted therapies to mitigate the impact of viral infections on chronic lung diseases.

## 6. Inflammatory Mediators

During viral infections, CLD leads to increased inflammation, impaired immune response, and tissue remodelling. Rhinoviruses, for instance, induce the release of IL-33, which promotes Th2 cytokine production and airway inflammation in asthma [103]. They also activate TLRs on airway epithelial cells, escalating bronchial inflammation and hyperresponsiveness [104]. Influenza viruses provoke elevated levels of pro-inflammatory cytokines, such as IL-6, TNF-α, and interferons, worsening airway inflammation and airflow limitation in COPD and asthma [105].

These viral infections trigger a cascade of immune responses. Viral replication in airway or lung cells leads to cytokine production, mucin production, and NO release, all of which are therapeutic targets in COPD. This cascade progresses to inflammation, cell damage, mucus hyperresponsiveness, and increased susceptibility to secondary bacterial infections. These mechanisms underlie viral infection-induced respiratory diseases such as pneumonia, ARDS, bronchitis, COPD exacerbations, bronchial asthma, and diffuse pan bronchiolitis (DPB) [106].

Inhaled corticosteroids (ICSs), long-acting muscarinic antagonists (LAMAs), long-acting β2 agonists (LABAs), and mucolytic agents possess anti-inflammatory and immunomodulatory properties that can mitigate these effects. Influenza infection, for example, stimulates nasal mucosal progenitor cells to recruit mature dendritic cells, enhancing the innate immune response and leading to lung fibrosis by producing transforming growth factor (TGF)-β [107]. Rhinovirus (RV) infection activates NF-κB and TLRs in bronchial epithelial cells, triggering the release of pro-inflammatory mediators such as IL-1β, IL-6, IL-8, RANTES, IP-10, and GM-CSF [108]. Additionally, RV infection induces mucin production through ATP release from infected cells via purinergic P2 receptors or the activation of SPDEF-regulated genes [109].

Inflammatory mediators like cytokines have various roles in exacerbating respiratory diseases. For instance, IL-1 induces the production of other inflammatory cytokines and mediators, airway remodelling, eosinophilia, and mucus hypersecretion. IL-6 plays a central role in the cytokine storm seen in COVID-19 and influenza virus infections and is associated with severe disease outcomes [96]. IL-8 is a key neutrophil chemoattractant in COPD, bronchiectasis, and DPB, while IL-4 and IL-5 are linked to exacerbations in eosinophilic asthma. Viral infections also induce angiogenic and growth factors like VEGF, FGF, and TGF-β, contributing to airway remodelling, interstitial lung disease, and ARDS by enhancing fibroblast proliferation and vascular permeability. Reactive oxygen species (ROS) and nitrogen species further induce airway inflammation, remodelling, hyperresponsiveness, mucus secretion, and cell damage.

Viral infections also disrupt normal lung repair processes, leading to pathological remodelling and exacerbation of CLDs [110]. Post-viral injury, the lung attempts to restore tissue integrity through reparative processes. However, in CLD patients, this is often dysregulated, leading to abnormal tissue remodelling and fibrotic scarring [111,112]. In COPD, viral-induced exacerbations accelerate lung function decline and increase disease progression risk [113]. Viral pathogens stimulate matrix metalloproteinases (MMPs) and other proteolytic enzymes, degrading the extracellular matrix and disrupting alveolar integrity [114]. This inflammation also promotes fibroblast activation and differentiation into myofibroblasts, resulting in excessive collagen deposition and pulmonary fibrosis [115]. In asthma, viral infections impair epithelial barrier function and promote airway remodelling, including subepithelial fibrosis and smooth muscle hypertrophy, exacerbating respiratory symptoms and reducing lung function [116,117].In ILDs, viral-induced exacerbations drive progressive fibrotic remodelling of the lung parenchyma, leading to irreversible lung function loss and poor clinical outcomes [118]. Viral pathogens stimulate the release of pro-fibrotic cytokines, such as TGF-β and PDGF, further contributing to pulmonary fibrosis [119].

Recent research highlights the role of lung stem/progenitor cells and related models in lung repair. The airway epithelium comprises basal cells with self-renewing abilities, which differentiate into various luminal cells, including ciliated cells, secretory cells, and less common cell types like ionocytes and neuroendocrine cells [120]. A single-cell analysis revealed multiple subpopulations of basal cells, each with distinct roles in health and disease, as seen in COPD [121]. However, comprehensive knowledge regarding these subpopulations’ roles in health and disease remains limited. Following damage, luminal cells, such as club cells, may also aid in repair, as demonstrated in rodent injury models [122].

Clinical studies have explored the effects of mesenchymal cell-derived fibroblast growth factors (FGFs) in lung development and repair. Intravenous FGF7 (keratinocyte growth factor) administration in ARDS patients, however, showed no clinical benefit and, in some cases, worsened outcomes compared to a placebo [99]. Retinoids and retinoic acid receptor agonists have shown promise in animal studies, but translating these findings into effective treatments for patients remains challenging [123].Researchers are also investigating therapies targeting cellular senescence, as accelerated aging and increased senescent cells are linked to lung diseases like COPD and idiopathic pulmonary fibrosis (IPF). These cells are less effective at repairing damage and release pro-inflammatory substances, contributing to disease progression. While senotherapy (senolytics and senostatics) shows potential in animal models, clinical trials are still needed to assess their effectiveness in humans [124]. Some drugs tested in animal studies have already been approved for human use for other medical conditions, but clinical studies specifically targeting senescence in lung diseases are still pending [125].

## 7. Preventive Measures and Early Detection

In COPD, a type of CLD, the concept of early COPD should be understood as an earlier stage in the disease course that does not yet present with airway obstruction or typical clinical symptoms. This stage must be distinguished from mild or severe COPD, which are more advanced forms of the disease [126]. However, due to a lack of evidence, many studies have not clearly differentiated mild COPD from early COPD [127].

Recently, Martinez et al. defined early COPD in patients under 50 years of age with a smoking history of at least 10 pack-years, which is considered the minimum exposure needed for the disease’s development. This definition focuses solely on cigarette exposure, the primary cause of COPD, though other risk factors include occupational exposure, previous viral infections, and air pollution [128]. Notably, this definition does not rely on airway obstruction as a criterion, allowing early COPD to be divided into two subtypes: pre-COPD and preclinical COPD [129]. Pre-COPD refers to patients without airway obstruction (FEV/FVC ≥ 0.7) who are at risk for COPD, while preclinical COPD describes patients with airway obstruction (FEV1/FVC < 0.7) but no or only mild respiratory symptoms such as cough, sputum production, or dyspnoea.

Prevention and early detection of viral infections are critical strategies for managing CLDs like COPD, asthma, and ILDs, as these patients are particularly vulnerable to severe complications from respiratory viral illnesses [130]. Vaccination is a key preventive measure, with annual influenza vaccination strongly recommended for individuals with CLD due to their higher risk of exacerbations and complications from influenza [131,132]. Influenza can worsen underlying respiratory conditions, lead to secondary bacterial infections, and increase hospitalization and mortality rates in this population. Similarly, pneumococcal vaccination is essential for preventing invasive pneumococcal disease, pneumonia, and exacerbations in CLD patients [133]. *Streptococcus pneumoniae*, the bacterium responsible for pneumococcal infections, poses a serious threat to those with compromised respiratory function, causing severe respiratory infections and worsening lung conditions [134]. Pneumococcal vaccines boost the immune system by producing protective antibodies against *S. pneumoniae* strains, reducing the risk of related complications in CLD patients [135].

Routine viral screening during the respiratory disease season is crucial for early detection of viral pathogens and timely initiation of preventive and therapeutic interventions [136]. Respiratory viral panel testing, which uses polymerase chain reaction (PCR) assays and antigen detection kits, allows for rapid identification of circulating viral strains [137]. In healthcare settings, such screening helps implement infection control measures promptly, preventing nosocomial transmission of respiratory viruses and protecting vulnerable patients, including those with CLDs [138]. Moreover, routine viral screening aids surveillance efforts, monitoring viral activity and trends to inform public health interventions and vaccination strategies.

Despite ongoing research, there have been no significant advancements in disease-modifying treatments for COPD, with smoking cessation remaining the only known intervention to slow disease progression and improve survival rates. Given the burden of established COPD, early diagnosis and intervention are crucial for addressing this global health challenge. The preventive measures are underlined in Figure 2.

## 8. Preventive Strategies for Managing CLDs

Preventing viral infections is crucial for managing CLDs to reduce exacerbations and enhance overall respiratory health (Figure 2).

The role of vaccinations, including influenza, pneumococcal, and COVID-19 vaccines, in preventing viral infections and minimizing exacerbations in CLD patients is an important preventive strategies for CLD management [139]. Annual influenza vaccination is strongly recommended for all individuals with CLDs, including COPD, asthma, and ILDs. Influenza viruses pose a significant risk of exacerbations and complications in these patients, increasing morbidity and mortality. Pneumococcal vaccination is also advised to prevent invasive pneumococcal disease, pneumonia, and exacerbations. *Streptococcus pneumoniae*, the bacterium responsible for pneumococcal infections, poses a significant threat to those with compromised respiratory function, leading to severe respiratory infections and worsening of underlying lung conditions [140]. Pneumococcal vaccines stimulate the immune system to produce protective antibodies against *S. pneumoniae*, bolstering defenses and reducing the risk of complications in individuals with CLDs.

The COVID-19 pandemic has highlighted the importance of vaccination for protecting vulnerable populations, including CLD patients, from severe respiratory illness and complications. COVID-19 vaccines, such as mRNA vaccines (e.g., Pfizer-BioNTech, Moderna) and viral vector vaccines (e.g., Johnson & Johnson), have shown high efficacy in preventing infection and reducing severe disease, hospitalization, and death [141]. Vaccination against COVID-19 is strongly recommended for individuals with CLDs to mitigate virus spread and safeguard public health [142].

In addition to vaccinations, behavioral and lifestyle modifications are essential for reducing the risk of viral infections and exacerbations in CLD patients. Good hygiene practices, such as frequent handwashing, respiratory etiquette (e.g., covering coughs and sneezes), and avoiding close contact with sick individuals, are crucial for preventing respiratory viruses like influenza, respiratory syncytial virus (RSV), and COVID-19 [143]. Smoking cessation is another key preventive measure. Smoking is a major risk factor for the development and progression of COPD, asthma, and other respiratory conditions. Quitting smoking can significantly improve lung function and reduce the frequency and severity of exacerbations [144]. Healthcare providers should offer counselling and support services to promote smoking cessation and improve respiratory health. Additionally, maintaining a healthy diet, engaging in regular physical activity, and ensuring adequate sleep can strengthen the immune system and reduce the risk of viral infections in CLD patients [145].

Exacerbations of COPD are often caused by infections, with various bacterial and viral species identified as culprits. Sputum samples from patients experiencing exacerbations frequently contain bacteria such as *Haemophilus influenzae*, *Streptococcus pneumoniae*, *Moraxella catarrhalis*, Enterobacteriaceae, and *Pseudomonas* species;viruses like rhinoviruses, influenza, parainfluenza, respiratory syncytial virus, and coronaviruses; and atypical organisms like *Mycoplasma pneumoniae* [32]. Identifying specific pathogens can be challenging, as many harmful microorganisms present during exacerbations may also be found in stable periods [146]. Environmental factors, such as air pollution, also contribute to exacerbations. Hospitalization and urgent care visits for severe COPD exacerbations impact patient well-being and healthcare costs. Pneumonia, a common comorbidity in COPD, notably increases the risk of hospitalization in elderly patients compared to those without lung diseases.

## 9. Targeted Treatment Approaches

Antiviral therapies and anti-inflammatory treatments are essential to managing CLDs, especially during viral infection-induced exacerbations (Figure 2). This section reviews the efficacy of antiviral therapies and the use of anti-inflammatory treatments in CLD populations [147].

### 9.1. Antiviral Treatments

Antiviral therapies aim to inhibit viral replication and reduce viral load, thereby lessening the severity and duration of respiratory viral infections. For individuals with CLD, including COPD and asthma, viral-induced exacerbations can accelerate disease progression and increase morbidity and mortality. Therefore, antiviral treatments targeting specific respiratory viruses are of considerable interest. Neuraminidase inhibitors, such as oseltamivir and zanamivir, are commonly used for influenza treatment [148]. These inhibitors block neuraminidase, an enzyme critical for viral replication and release from infected cells, thereby reducing viral spread and disease severity [149]. Although neuraminidase inhibitors have shown efficacy in the general population, their effectiveness in CLD patients remains uncertain [150].

Recent studies have examined the efficacy of neuraminidase inhibitors in CLD patients experiencing viral-induced exacerbations [151]. Some studies report modest benefits in symptom improvement and reduced healthcare utilization, while others indicate limited efficacy, especially in patients with severe or advanced disease [152]. The emergence of antiviral resistance also challenges the effectiveness of neuraminidase inhibitors [153]. Besides neuraminidase inhibitors, other antiviral agents, such as nucleoside analogs and protease inhibitors, are being explored for treating respiratory viral infections in CLD patients. However, more research is needed to determine optimal antiviral strategies and identify new therapeutic targets for managing viral-induced exacerbations [154].

Antiviral drug therapy began with the development of iododeoxyuridine (IDU) in 1959, originally intended as an anticancer medication. In the late 1970s, the introduction of acyclovir (ACV) and its prodrugs marked a significant advancement in the treatment of viral infections. ACV targets HSV-infected cells through phosphorylation by a viral enzyme, demonstrating low toxicity and high specificity [155]. ACV inhalation has also been reported to have antiasthma activity, improving the airway obstruction and suppressing the eosinophil influx in bronchoalveolar lavage fluid [156]. Thus, ACV may be considered for antiviral treatment in CLD patients. A single-center randomized double-blind placebo-controlled trial showed that Valaciclovir, a nucleoside prodrug of acyclovir, suppressedoral Epstein-Barr virus (EBV) replication in moderate to severe COPD [157]. In comparison to oseltamivir monotherapy, the combined effect of ribavirin, amantadine, and oseltamivir against influenza was shown to be more effective [158].

Other substances that suppress the host’s immune response to regulate viral infections have also been studied, in addition to antivirals that function by disrupting the replication cycle. Interferons activate the innate immune response against viral infections through cytokines. In summary, interferons have the ability to provide resistance against viral infection, or an antiviral state, to both healthy and infected cells. Infected cells secrete interferons, which bind to cell receptors, particularly on neighbouring cells, and activate transcription of genes encoding antiviral proteins, like ribonucleases, thereby blocking viral replication [159,160]. Clinical studies showed that a reduction in interferon production increased the susceptibility to virus infections and increased exacerbation frequency in COPD patients;additionally, inhaled IFN-β therapy benefited asthmatic patients with viral infections [161]. Further, the antiviral imiquimod is an immune response modifier, which functions as an antagonist to toll-like receptors. The recognition of conserved regions of pathogens and the induction of cytokine production by toll-like receptors play a critical role in the immune response against infections. Imiquimod is regarded as a potent antiviral drug to increase the interferon expression and reduce the ACE2 and pro-inflammatory response of human bronchial epithelium in asthma [162].

### 9.2. Anti-Inflammatory Treatments

Anti-inflammatory treatments are crucial for managing exacerbations of CLD induced by viral infections. These exacerbations are marked by increased airway inflammation, leading to bronchoconstriction, mucus hypersecretion, and airflow limitation [163]. Anti-inflammatory therapies are essential for alleviating these symptoms and improving clinical outcomes [164].

Corticosteroids, such as prednisone and dexamethasone, are commonly used to manage exacerbations in CLD patients. They work by suppressing pro-inflammatory cytokines, inhibiting inflammatory cell migration, and reducing airway oedema and mucus production [165]. While effective in reducing airway inflammation and improving lung function, long-term use of corticosteroids is associated with adverse effects, including immunosuppression, osteoporosis, and metabolic disturbances [166].

In addition to systemic corticosteroids, ICSs are used as maintenance therapy to prevent exacerbations and control airway inflammation in asthma and COPD patients [167]. During viral-induced exacerbations, high-dose ICSs, alone or in combination with long-acting beta-agonists (LABAs), may help reduce airway inflammation and improve symptom control [35]. However, the effectiveness of ICSs in managing viral-induced exacerbations varies and should be guided by individual patient factors and disease severity [168].

Other anti-inflammatory treatments, such as phosphodiesterase-4 (PDE4) inhibitors and macrolide antibiotics, are being explored for their role in managing CLD exacerbations [169]. PDE4 inhibitors, like roflumilast, reduce pro-inflammatory mediators by inhibiting cyclic adenosine monophosphate (cAMP) breakdown [170]. Macrolide antibiotics, such as azithromycin, have anti-inflammatory properties independent of their antimicrobial effects and may improve clinical outcomes in CLD patients with exacerbations [171].

## 10. Diagnostic and Management Challenges

COPD is a progressive condition characterized by airflow limitation and persistent respiratory symptoms, typically caused by exposure to harmful particles or gases, primarily cigarette smoke [172]. COPD exacerbations, which involve acute worsening of symptoms, significantly impact patient outcomes, leading to increased morbidity, mortality, and healthcare utilization. Viral infections contribute to approximately 50% of COPD exacerbations, with rhinovirus being the most common pathogen [173]. Diagnosing viral exacerbations is challenging due to the overlap in clinical presentations with bacterial exacerbations and limitations in diagnostic methods. Bacterial exacerbations are often associated with purulent sputum and systemic signs like fever and elevated white blood cell count, andviral exacerbations may present similarly [174].

Accurately differentiating viral from bacterial exacerbations is critical for appropriate treatment and to avoid unnecessary antibiotic use, which can reduce the risk of antimicrobial resistance and adverse effects. Diagnostic modalities, including sputum cultures, blood tests, and imaging, have limitations in sensitivity, specificity, and turnaround time, especially in outpatient settings [175]. Therefore, clinical judgment and symptom assessment are often relied upon. Improving diagnostic accuracy, such as through rapid point-of-care tests for respiratory viruses, is essential for timely identification and targeted treatment [176].

Management of viral-induced exacerbations of CLD involves a multifaceted approach to alleviate symptoms, reduce inflammation, and prevent disease progression [177]. Antiviral therapies, like neuraminidase inhibitors, are crucial, particularly for severe or advanced lung disease cases. While effective in reducing influenza severity, their effectiveness in CLD patients is still under investigation [178]. In addition to antiviral agents, adjustments in maintenance therapies, including bronchodilators and corticosteroids, may be necessary. Short-acting bronchodilators, such as albuterol, offer rapid relief from bronchospasms, while systemic corticosteroids help reduce inflammation and improve lung function [179]. High-dose inhaled corticosteroids and short courses of oral corticosteroids may be needed to manage viral-induced exacerbations in asthma patients.

Pulmonary rehabilitation (PR) has shown positive impacts on exercise capacity, lung function, respiratory muscle strength, and quality of life in chronic respiratory disease patients [179]. Despite evidence linking physical inactivity to poor outcomes in COPD, increased exercise tolerance alone may not lead to more physical activity. Troosters et al. found that PR was three times more effective than bronchodilators in increasing endurance during exercise testing [180]. Figure 2 summarizes the diagnostic and management challenges of CLD during viral infections.

## 11. Future Directions

Emerging antiviral drugs and biologic therapies offer promising avenues for reducing the impact of viral infections on CLD and improving respiratory health outcomes [181]. New antiviral drugs targeting specific pathogens, such as respiratory syncytial virus (RSV), rhinovirus, and influenza, are under development and may provide novel treatment options for CLD patients experiencing viral-induced exacerbations [182]. These drugs aim to inhibit viral replication, reduce viral load, and lessen disease severity.

Biologic therapies targeting key inflammatory pathways involved in viral-induced exacerbations, such as interleukin (IL)-4, IL-5, and IL-13 in asthma, and tumor necrosis factor-alpha (TNF-α) and IL-1 in COPD, show potential for modulating airway inflammation and reducing respiratory symptoms [183]. By targeting specific inflammatory mediators and immune cells, these therapies offer a targeted approach to managing CLDs and may complement existing treatments, including bronchodilators and corticosteroids.

A significant research gap in developing targeted therapies and vaccines for CLD populations is the need for individualization and optimization based on diverse clinical phenotypes and underlying pathophysiology [184]. CLDs, including COPD, asthma, and ILDs, encompass heterogeneous disease entities with varying severity, inflammatory profiles, and treatment responses. Current therapies often use a generalized approach, which may result in suboptimal outcomes. Personalized medicine approaches that consider individual patient characteristics, such as disease phenotype, inflammatory profile, and treatment response, are necessary. Identifying biomarkers and clinical predictors to stratify patients into distinct subgroups can enhance treatment efficacy and outcomes [185].

Additionally, developing targeted therapies and vaccines tailored to specific CLD populations, addressing their unique pathophysiological mechanisms, is essential. Novel biologic therapies targeting specific inflammatory pathways or immune cell subsets may offer more precise treatment options [186]. Optimizing vaccine formulations, dosing regimens, and delivery methods to enhance efficacy in individuals with impaired immune function or altered vaccine responsiveness are desired.

A detailed overview of the mechanisms of disease exacerbation, immune dysregulation, and managing viral infections in CLDs is depicted in Table 1.

## 12. Conclusions

Viral pathogens frequently exacerbate underlying respiratory conditions, significantly increasing morbidity and mortality, reflecting the complex relationship between viral infections and CLDs. This review examined how viral infections influence disease progression and exacerbations and discussed strategies to mitigate their impact. The bidirectional relationship between ILDs, asthma, COPD, and viral infections illustrates how pre-existing lung conditions can exacerbate viral illnesses and how viral infections can worsen underlying lung conditions. Preventive strategies, including vaccinations, behavioral modifications, and early detection through routine viral screening, play a crucial role in reducing the burden of viral infections and improving respiratory health in CLD patients.

Vaccinations against influenza, pneumococcal pathogens, and COVID-19 are critical for preventing infections and reducing exacerbations in vulnerable populations. We discussed the efficacy of antiviral therapies, anti-inflammatory treatments, and targeted biologic therapies in managing viral-induced exacerbations and mitigating the impact of viral infections on CLDs. Future research should focus on personalized therapies and vaccines tailored to specific CLD populations and optimizing treatment strategies based on individual disease characteristics [193]. By integrating preventive measures, targeted treatments, and personalized approaches, healthcare providers can enhance clinical outcomes and improve respiratory health in CLD patients facing viral challenges.

## Figures and Tables

**Figure 1 microorganisms-12-02030-f001:**
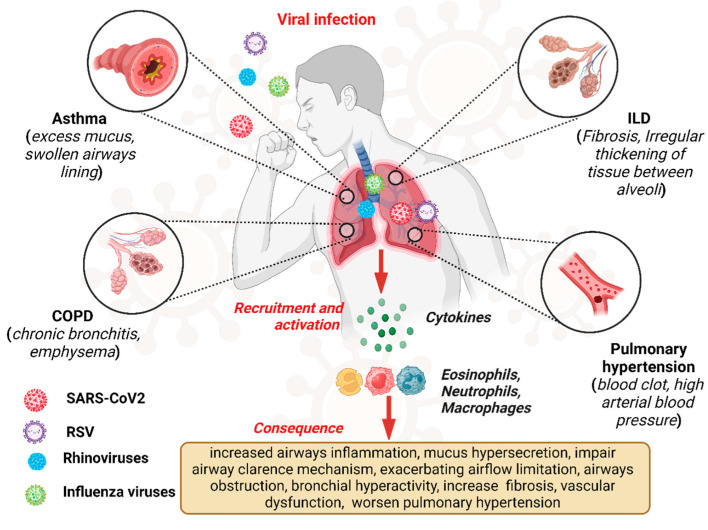
Overview of viral infections, pathway, and consequences in chronic lung disease.“Created in BioRender. Pande, B. (2024) BioRender.com/v50i654 (accessed on 25 August 2024)”.

**Figure 2 microorganisms-12-02030-f002:**
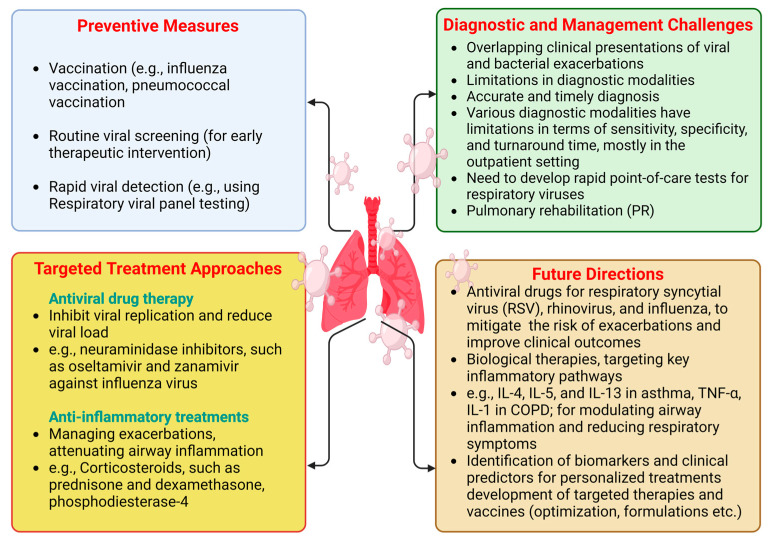
Preventive measures, targeted treatment approaches, diagnostic and management challenges, and future directions for CLD patients during viral infection periods. “Created in BioRender. Pande, B. (2024) BioRender.com/o38j486 (accessed on 25 August 2024)”.

**Table 1 microorganisms-12-02030-t001:** Summary of the mechanisms of disease exacerbation, immune dysregulation, inflammatory pathways, biomarkers, and therapeutic strategies for managing viral infections in CLDs [187,188,189,190,191,192].

Disease/ Characteristic	COPD	Asthma	Interstitial Lung Disease	Pulmonary Hypertension
Disease Exacerbation Mechanism	Viral infections increase serum IL-6 and fibrinogen, intensifying inflammation and tissue damage.	Viral infections trigger eosinophilic inflammation, increasing Th2 cytokines (IL-4, IL-5, IL-13).	Viral-induced dysregulation leads to pro-fibrotic cytokine release (TGF-β, PDGF), causing fibrosis.	Viral infections exacerbate vascular remodeling, causing increased pulmonary vascular resistance and pressure.
Immune Dysregulation	Pollutants and cigarette smoke attract CD8/CD4 T cells and neutrophils, leading to tissue remodeling and emphysema.	Compromised epithelial barriers and mucociliary clearance impaired cytokine signalling worsened viral outcomes	Dysregulated immune responses cause progressive fibrotic remodelling.	Altered immune cell function and increased pro-inflammatory cytokines exacerbate vascular remodelling.
Inflammatory Pathways	Pro-inflammatory cytokines (IL-6, IL-8, TNF-α) drive neutrophil and macrophage activation, worsening inflammation.	IL-33 releaseslead to Th2 cytokine production and airway inflammation.	Pro-fibrotic cytokines (TGF-β, PDGF) cause myofibroblast activation, collagen deposition, and fibrosis.	Inflammatory mediators (IL-6, IL-8) contribute to pulmonary vascular remodelling and right ventricular strain.
IL-6 as a Biomarker	Elevated IL-6 correlates with severe disease and exacerbations, making it a potential therapeutic target.	IL-6 is a key cytokine in the inflammatory response and a potential target in severe asthma exacerbations.	IL-6 contributes to fibrosis progression and may serve as a biomarker for disease severity.	Elevated IL-6 levels linked to PH progression and severity; potential biomarker for disease monitoring
Inflammatory Mediators	IL-1, IL-6, and IL-8 exacerbate respiratory inflammation, leading to tissue damage and mucus hypersecretion.	Similar pathways between IL-1 and IL-33 induce inflammation and increase mucus production and airway constriction.	TGF-β and VEGF contribute to airway remodelling and fibrosis, leading to worsening lung function.	Inflammatory mediators induce vascular inflammation, promoting progression to advanced PH.
Preventive Measures	Annual influenza and pneumococcal vaccination are crucial to prevent exacerbations.	Early detection and vaccination	Early vaccination to prevent viral-induced exacerbations and fibrotic progression	Vaccination to reduce the risk of viral infections exacerbating PH and overall lung function
Targeted Treatment Approaches	Antivirals like oseltamivir; anti-inflammatories like corticosteroids and PDE4 inhibitors	ICS and corticosteroids to control inflammation; antivirals for reducing viral load	Antifibrotic and anti-inflammatory treatments to manage exacerbations and progression	Antiviral and anti-inflammatory therapies to mitigate vascular inflammation and PH progression
Diagnostic Challenges	Difficulty distinguishing viral from bacterial exacerbations; need for rapid diagnostic tests	Challenges distinguishing viral from bacterial exacerbations; rapid and accurate viral identification needed	Challenges differentiating viral-induced exacerbations; reliance on advanced diagnostics	Complicated diagnosis due to overlapping symptoms; importance of accurate viral detection
Preventive Strategies	Early diagnosis, vaccination, and targeted treatment to reduce exacerbation frequency and severity	Focus on early detection, vaccination, and maintaining airway function through preventive strategies	Preventive measures aimed at reducing fibrosis progression and maintaining lung function	Emphasis on early diagnosis and treatment to prevent exacerbation and progression to severe PH

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
