# Peer review of "Interplay between Lung Diseases and Viral Infections: A Comprehensive Review"

_microorganisms, 2024, doi:10.3390/microorganisms12102030_

Round 1

Reviewer 1 Report

Comments and Suggestions for Authors

     The authors explored the influence of pre-existing lung conditions on susceptibility, severity, and outcomes of viral infections, including chronic obstructive pulmonary disease, asthma, and interstitial lung diseases. The paper also examined the mechanisms that viral infections exacerbated and accelerated the progression of lung disease, explaining the intricate relationship between chronic lung diseases and viral infections. I think it's a good review, but before publishing, the author needs to make the following minor revisions.

1. Whether the keyword “mechanism” should be changed? It has a somewhat broad meaning.

2. There are two “3.2” in the text, which need to be revised.

3. First-line indentation should be consistent, such as lines 33, 87, 121, 131, 141, 171, 179 etc. The Authors should scrutinize the full review.

4. What does “Sentence missing for this citation” mean in line 456?

5. Whether the title of part 8 should be changed to “Anti-viral Treatments” in order to correspond to part 9 “Anti-Inflammatory Treatments”?

6. What about part 8 and 9 together as “Targeted Treatment Approaches” and then divided into “Anti-viral Treatments” and “Anti-Inflammatory Treatments”?

7. The distance between lines 669 and 673 needs to be revised.

8. Line 696 should be amended to “function or altered vaccine responsiveness., and a blank line is required between lines 696 and 697.

9. Whether line 698 should be changed to “managing viral infections in CLD is depicted in Table 1.?

10. The figure note in Table 1 needs to be aligned with the table.

11. Lines 715, 716, 717, 731 should be revised.

Author Response

Comments and Suggestions for Authors

     The authors explored the influence of pre-existing lung conditions on susceptibility, severity, and outcomes of viral infections, including chronic obstructive pulmonary disease, asthma, and interstitial lung diseases. The paper also examined the mechanisms that viral infections exacerbated and accelerated the progression of lung disease, explaining the intricate relationship between chronic lung diseases and viral infections. I think it's a good review, but before publishing, the author needs to make the following minor revisions.

  1. Whether the keyword “mechanism” should be changed? It has a somewhat broad meaning.

Compliance/Reply: Thanks for the valuable comment. We have replaced “mechanism” with “interaction” and “exacerbation”.

  1. There are two “3.2” in the text, which need to be revised.

Compliance/Reply: Thanks for your observation. We have revised the said portion.

  1. First-line indentation should be consistent, such as lines 33, 87, 121, 131, 141, 171, 179 etc. The Authors should scrutinize the full review.

Compliance/Reply: Thanks for your observation. We corrected the indentation throughout the text.

  1. What does “Sentence missing for this citation” mean in line 456?

Compliance/Reply: Thanks for your scrutiny. We have corrected the mistake.

  1. Whether the title of part 8 should be changed to “Anti-viral Treatments” in order to correspond to part 9 “Anti-Inflammatory Treatments”?

Compliance/Reply: Thanks for the valuable suggestion. We have replaced incorporated “Antiviral Treatments” as subheading (8.1) under heading Targeted Treatment Approaches and “Anti-Inflammatory Treatments” as other subheading (8.2).

  1. What about part 8 and 9 together as “Targeted Treatment Approaches” and then divided into “Anti-viral Treatments” and “Anti-Inflammatory Treatments”?

Compliance/Reply: Thanks for the critical observation. We have revised the headings, subheadings and sequence of numbering.

  1. The distance between lines 669 and 673 needs to be revised.

Compliance/Reply: Thanks for the comment. We revised the spacing problem between 669 and 673.

  1. Line 696 should be amended to “function or altered vaccine responsiveness.”, and a blank line is required between lines 696 and 697.

Compliance/Reply: Thank you for your comments. We have amended the sentence and given space between lines 696 and 697.

  1. Whether line 698 should be changed to “managing viral infections in CLD is depicted in Table 1.”?

Compliance/Reply: Thanks for the valuable comment. Since the table related to “themechanisms of disease exacerbation, immune dysregulation for managing viral infections in CLD”, so we retain the sentence and as per suggestion remove the bracket before after citation of Table 1.

  1. The figure note in Table 1 needs to be aligned with the table.

Compliance/Reply: Thanks for the comment. We have adjusted the Table title according to table.  Since figures are within text the alignment of figure legends are according to text. Table 1 giving independent information and in different page so we have tried to adjust within the space.

  1. Lines 715, 716, 717, 731 should be revised.

Compliance/Reply: Thanks for the valuable comment. We have revised the lines 715, 716, 717, 731.

Thank you very much for the great review and improving our manuscript.

Best Regards

Reviewer 2 Report

Comments and Suggestions for Authors

Suri et al. in their review attempted to review the current state of the Interplay Between Lung Diseases and Viral Infections. As presented, the review is not a finished manuscript, but rather a draft. The authors use different formatting in the text (e.g. red line, spacing); chapter numbers are confused. In absctract, a sentence begins with a small letter. Some of the references are mixed up and not properly spaced in the text (e.g., reference 2 and 169).  This manuscript should be checked and formatted more thoroughly.

Major revision

In the introduction, more recent references are needed to support the thesis that viral infections are the most frequent trigger of CLD exacerbations.

It is also unclear where the thesis that infection is the cause of CLD exacerbations comes from, as the authors themselves describe that antiviral therapy has almost no effect on CLD exacerbations, unlike anti-inflammatory therapy. Could the inflammatory process and the immune status of the patient be the cause of the exacerbation after all?

An additional weakness is the frequent repetition of the same ideas in different paragraphs of the review, e.g. paragraphs 2.1-2.4 heavily duplicate paragraphs 3, 3.2 and 3.2. 

It is not clear why the authors provide information about COVID-19 in paragraph 5. What does it have to do with CLD?

It is also not clear why 8 describes the mechanisms of action of quite old and well-known antiviral drugs, some of which are not associated with CLD.

Author Response

Reviewer 2

Comment:  Suri et al. in their review attempted to review the current state of the Interplay Between Lung Diseases and Viral Infections. As presented, the review is not a finished manuscript, but rather a draft. The authors use different formatting in the text (e.g. red line, spacing); chapter numbers are confused. In absctract, a sentence begins with a small letter. Some of the references are mixed up and not properly spaced in the text (e.g., reference 2 and 169).  This manuscript should be checked and formatted more thoroughly.

Compliance/Reply: Thanks for the precious comments. We have corrected the errors in the MS such has formatting pattern, capitalized the first letter in a sentence in the abstract, properly spaced references and numbering of sections.

Major revision

Comment: In the introduction, more recent references are needed to support the thesis that viral infections are the most frequent trigger of CLD exacerbations.

Compliance/Reply: Thanks for the valuable suggestions. There are fifteen recent articles (years 2021-2024) cited in introduction, specifically reference 8,9,10, 11,14,15,16. We have added a few relevant references in MS where needed.

Comment: It is also unclear where the thesis that infection is the cause of CLD exacerbations comes from, as the authors themselves describe that antiviral therapy has almost no effect on CLD exacerbations, unlike anti-inflammatory therapy. Could the inflammatory process and the immune status of the patient be the cause of the exacerbation after all?

Compliance/Reply: Thanks for the valuable comments. Viral infection worsens the exacerbation rather than the cause of the exacerbation. Many antiviral drugs have tested that we have stated in antiviral section. Some did not showed effect while some have been linked with the alleviation of symptoms that we have mentioned in antiviral section. We have added new references regarding antiviral drugs., We incorporated  some of the drugs were studied in CLD. There are drugs acts as both antiviral as well as anti-inflammatory such as Imiquimod. Yes agree that the inflammatory process and the immune status of the patient be the cause of the exacerbation, nonetheless viral infections acts as trigger for the inflammatory process and the immune status worsening the symptoms leading to intensification of exacerbation.

Comment: An additional weakness is the frequent repetition of the same ideas in different paragraphs of the review, e.g. paragraphs 2.1-2.4 heavily duplicate paragraphs 3, 3.2 and 3.2. 

Compliance/Reply: Thanks for the critical observation. Section 2 is related to viral infection as an agent that triggers the exacerbation in different types of CLD, with similar pathways. Therefore, there are repetitions (specifically the names of the viruses and the infection site) in some portions. Section 3 is the mechanism of exacerbation in CLD given in common, then in immune dyregulation and Inflammatory Pathways and Viral Exacerbation which have certain similarities in mechanism. We have amended some of the repetition.

Comment: It is not clear why the authors provide information about COVID-19 in paragraph 5. What does it have to do with CLD?

Compliance/Reply: Thanks for the valuable comment. COVID-19 is caused by SARS-CoV-2  virus that have worsen the chronic lung diseases such as have been correlated with the progression of fibrosis in ILD patients (Fukihara and Kondoh, 2023, now incorporated in ILD section 2.3) or linked to Pulmonary Hypertension (PH) in COPD patients. It is well studied that patients with CLD are at higher risk of severe infection specifically immunocompromised patients like ILD (Calver et al. 2023). Therefore, we have explained the role of IL-6 as a Biomarker and Therapeutic Target in section 5.

Reference:

Calver JF, Fabbri L, May J, Jenkins RG. COVID-19 in Patients with Chronic Lung Disease. Clin Chest Med. 2023 Jun;44(2):385-393. doi: 10.1016/j.ccm.2022.11.013. Epub 2022 Nov 22. PMID: 37085227; PMCID: PMC9678841.

Comment: It is also not clear why 8 describes the mechanisms of action of quite old and well-known antiviral drugs, some of which are not associated with CLD.

Compliance/Reply: Thanks for the important comments. We have revised the antiviral section. We have added the recent article and removed the antiviral drugs which are not directly associated with the CLD and which could be considered to treat chronic lung diseases.

Thank you very much for the great review and improving our manuscript.

Best Regards